# Multifunctional Three-Dimensional Printed Copper Loaded Calcium Phosphate Scaffolds for Bone Regeneration

**DOI:** 10.3390/ph16030352

**Published:** 2023-02-25

**Authors:** Amit Pillai, Jaidev Chakka, Niloofar Heshmathi, Yu Zhang, Faez Alkadi, Mohammed Maniruzzaman

**Affiliations:** PharmE3D Lab, Division of Molecular Pharmaceutics, College of Pharmacy, The University of Texas at Austin, Austin, TX 78712, USA

**Keywords:** 3D printing, copper nanoparticles, calcium phosphate cement, bone regeneration, bone angiogenesis, bone antimicrobial activity

## Abstract

Bone regeneration using inorganic nanoparticles is a robust and safe approach. In this paper, copper nanoparticles (Cu NPs) loaded with calcium phosphate scaffolds were studied for their bone regeneration potential in vitro. The pneumatic extrusion method of 3D printing was employed to prepare calcium phosphate cement (CPC) and copper loaded CPC scaffolds with varying wt% of copper nanoparticles. A new aliphatic compound Kollisolv MCT 70 was used to ensure the uniform mixing of copper nanoparticles with CPC matrix. The printed scaffolds were studied for physico-chemical characterization for surface morphology, pore size, wettability, XRD, and FTIR. The copper ion release was studied in phosphate buffer saline at pH 7.4. The in vitro cell culture studies for the scaffolds were performed using human mesenchymal stem cells (hMSCs). The cell proliferation study in CPC-Cu scaffolds showed significant cell growth compared to CPC. The CPC-Cu scaffolds showed improved alkaline phosphatase activity and angiogenic potential compared to CPC. The CPC-Cu scaffolds showed significant concentration dependent antibacterial activity in *Staphylococcus aureus*. Overall, the CPC scaffolds loaded with 1 wt% Cu NPs showed improved activity compared to other CPC-Cu and CPC scaffolds. The results showed that copper has improved the osteogenic, angiogenic and antibacterial properties of CPC scaffolds, facilitating better bone regeneration in vitro.

## 1. Introduction

Bone defects resulting from trauma or tumor resection result in the development of large fractures and can often cause insufficient innate healing, leading to a condition called non-union fractures. The gaps left in the bone will cause several infections such as osteomyelitis [1,2,3]. Autografts are the gold standard for treating these defects, but they suffer from the issue of being limited in supply [4]. Synthetic options such as calcium phosphate cements (CPCs) are a commonly used alternative to restore the anatomy as well as promoting bone ingrowth. CPCs are composed of a reactive calcium phosphate powder which, when in contact with a liquid carrier such as water, undergoes a setting reaction involving dissolution-precipitation to form hydroxyapatite (HA) which is the primary mineral in the bone. The inherent osteoconductive and osteoinductive properties of CPC make them a great alternative to autografts [5]. CPC is malleable in nature and can be injected into the defect site to perfectly fill the defect geometry [6]. Traditional cements are often treated as ticking time bombs as they get hardened before placement in the body and hence need to be injected prior to setting [7]. For this reason, pasty calcium phosphate cements have garnered widespread interest as they do not require setting prior to application into the body as well as their ability to remain stable for extended periods of time [8,9,10].

The formation of new blood vessels (i.e., angiogenesis) is essential for osteoblast cell survival and complete osteogenesis. The CPC that is directly injected to the defect site has inherent porosity which is often insufficient for angiogenesis [11]. A pore size ranging from 300–500 µm is considered essential for angiogenesis [12,13]. To achieve the desired porosity, 3D printing is the optimal technique. 3D printing is an additive manufacturing process which builds scaffolds in a layer-by-layer fashion by extruding highly viscous material through a needle using the pneumatic extrusion method [14,15]. As per the material, the plotted structures often require a post-processing step to stabilize the printed structures. Using this process developed by Landers and Mulhaupt, CPCs can be molded into predefined shapes to match the defect geometry as well as create scaffolds with pore sizes required for angiogenesis [16]. Further, Lode et al., used this process for the fabrication of porous scaffolds using pasty CPC under mild conditions. They successfully printed CPC scaffolds using the extrudable malleable paste and set them using immersion in water at 37 °C [17].

Copper is an element that has shown the potential to enhance the proliferation of endothelial cells and in turn promote angiogenesis in vivo [18,19]. Furthermore, copper present in the wound tissue is required for mediating the engendering of free radicals in regenerating tissues [20]. It is also pivotal for bone metabolism as well as turnover in the growth of bone tissue [12]. Barralet et al. harnessed this potential to 3D print degradable microporous CPC scaffolds with copper ions absorbed onto it to investigate the angiogenic potential of copper sulfate. These scaffolds were implanted intraperitoneally into rats, and it was found that copper did in fact improve vascularization as well as wound healing in the implanted regions [12]. Copper has also been used as an antimicrobial agent. The biocidal effect of copper has been widely studied and it has been proven with a myriad of pathogens, bacteria, fungi, as well as viruses. Further copper is effective against methicillin resistant *Staphylococcus aureus* (MRSA) [17,21,22,23].

In this study, we describe for the first time the addition of copper nanoparticles into pasty CPC to create 3D plottable calcium phosphate cements with osteogenic, angiogenic, and antimicrobial properties. This study is also the first attempt to use a novel oil phase composition to incorporate copper without losing the inherent extrudability of the paste.

## 2. Results

### 2.1. Injectability

The extrudability of CPC paste was pivotal to understanding the capability of the CPC and Cu loaded CPC materials to be used for 3D printing. Although the addition of Cu did increase the peak extrusion force needed to extrude the cement from the syringe, the paste did retain its extrudability. It was initially observed that the addition of Cu nanoparticles alone at high concentrations into the CPC had impeded the extrudability. To overcome this problem, a liquid phase (Kollisolv MCT 70, Polysorbate 80 and Lutensit A-EP) in proportion to the Cu-NP, which is 4:1 solid phase to liquid phase, was added to the CPC to maintain the extrudability of the paste. This modification facilitated the better 3D printing of Cu loaded CPC scaffolds with different wt% (0.1, 0.5, 1, 3, and 5). The extrusion force for the formulations was shown in Table 1. The force increased with the addition of Cu nanoparticles in CPC from 3090 ± 41 g to a maximum of 4524 ± 150 g.

### 2.2. 3D Printing

The CPC and copper loaded CPC were 3D printed using a pneumatic extrusion 3D bioprinter using print parameters at different wt% of copper nanoparticles as reported in Table 1.

### 2.3. Scaffold Characterization

#### 2.3.1. Optical Microscopy

The optical microscopy images showed the color change of the scaffolds with increasing amounts of copper nanoparticles as shown in Figure 1. The strut diameter and the pore size calculated using ImageJ were reported for all the scaffolds in Table 1.

#### 2.3.2. Scanning Electron Microscopy (SEM)

The SEM images of Cu nanoparticles (Appendix A) and CaP-Cu scaffolds (Figure 1) at 100× showed a smooth surface morphology with clear pores. The zig-zag overlay is visible with struts printed with the first layer at 0 degrees and the second layer at 90 degrees orientation. The images at 5000× showed the typical CPC scaffold morphology with a micropore surface. As the setting was carried out in a humidified chamber, the typical flower-like morphology of the hydroxyapatite (HaP) is not evident [24]. There are no aggregates of Cu nanoparticles on the surface of Cu loaded CPC scaffolds, indicating thorough and uniform distribution of the particles.

#### 2.3.3. Contact Angle Measurement

The uniform distribution of particles has shown an immediate effect on the contact angle with an almost 50% reduction in value between CPC and CPC-Cu 0.1 scaffolds. Figure 1 shows that the CPC scaffold has a contact angle of 75 ± 9° and all the CPC-Cu scaffold surfaces except for CPC-Cu 0.1 have a contact angle ≤15° which means those surfaces are hydrophilic [25,26]. The surface texture of the copper substrate significantly affected the wetting properties of the CPC scaffold. The change in contact angle is very minimal between the scaffolds CPC-Cu 0.5, 1, and 3 (Table 1). The CPC-Cu 5 showed the contact angle value of 3 ± 1° which shows the high wettability of the scaffolds. The wettability of the scaffold reflects the cell attachment and cell proliferation properties [27].

#### 2.3.4. Fourier Transform Infrared Spectroscopy

Considering that the Cu-NPs were mixed using a mortar and pestle, no interactions were expected initially. However, FTIR spectra (Figure 2A) were obtained after the setting reaction which could cause a change in the pH of the local milieu [28]. The peak at 880 cm^−1^ is due to C-H group bending vibration. The asymmetric stretching vibrations of the C-O-C bond are identified at 1035 cm^−1^ and 1150 cm^−1^. The C=O symmetrical stretch gives a sharp peak at 1740 cm ^−1^, while the peaks at 2930–2940 cm^−1^ and 2850–2870 cm^−1^ are, respectively, the characteristic bands of the symmetric and asymmetric vibrations of methylene groups from CPC. From the FTIR spectrum, it is evident that no interactions were observed after setting either, concluding that the setting reaction does not cause any interactions between the Cu-NPs and CPC. The bands obtained at 1585 and 1632 cm^−1^ correspond to carbonyl C=O stretching bands. The C–H stretching bands are observed in the region of 3300–2800 cm^−1^. Moreover, the broad band centered at 3413 and 3392 cm^−1^ is attributed to the stretching and bending vibrations of absorbed water and surface −OHs, confirming the absorption band of Cu nanoparticles present on the spectra [29,30].

#### 2.3.5. Powder X-ray Diffraction (XRD)

As per the X-Ray diffraction patterns (Figure 2B) of the samples post 3 days in a humidity chamber for setting, it is evident that the HA peaks remain to be observed between 0° ≤ 2θ ≤ 80° even in the Cu loaded samples without additional CaO phases. The characteristic HA peaks (JCPDS 9-0432) (hkl values: (002), (211), and (202)) at 2θ of 25.8, 32.1, and 35.0 are observed in all samples [31,32]. The representative copper peaks at 2θ of 43.6 and 50.7 can be seen in all the Cu loaded samples [24]. Further, on increasing metallic copper loading, the copper XRD peaks increase in intensity.

#### 2.3.6. Copper Ion Release Study

The copper release for CPC-Cu scaffolds (Figure 3) evaluated at physiological pH conditions of 7.4 showed the copper ions released from the metallic copper nanoparticles. The release is concentration dependent between the scaffolds on day 1. There is no significant difference between the groups CPC-Cu 3 and CPC-Cu 5. The release for CPC-Cu 1 is lesser by day 7 and 14 compared to day 1. However, the release of Cu is higher in the CPC-Cu 1 group compared to CPC-Cu 0.1 and 0.5 groups which showed the concentration dependent release.

### 2.4. In Vitro Studies

#### 2.4.1. Cell Attachment

The SEM micrographs as well as fluorescence microscopy images for cell attachment (Figure 4A) analysis after 1 day show that hMSCs were in fact able to attach to the scaffolds. From visual observation, it was seen that there were a lower number of cells seen in scaffolds CPC-Cu 3 and CPC-Cu 5. A change in morphology was observed in all the copper loaded scaffolds from a linear, elongated structure to a circular morphology [33].

#### 2.4.2. Cell Proliferation

It can be seen from the MTT assay, that cell proliferation levels were similar in most of the Cu loaded scaffolds to the blank CPC for a period of seven days (Figure 4B). O.D. values were calculated by subtracting the blank value. It is interesting to note that over the period of a day, blank CPC scaffolds showed the least amount of hMSC proliferation. Furthermore, scaffolds CPC-Cu 0.1, 0.5, as well as 1 showed better hMSC viability as compared to the tissue culture polystyrene (TCPS) control. It can also be noted that scaffold CPC-Cu 3 and CPC-Cu 5 showed negative O.D. values that indicate significant cell death. On day 3, scaffold CPC-Cu 0.1 showed the best hMSC viability, however, did not show a significant difference compared to the blank. It should also be noted that CPC-Cu 1 showed improved viability whereas CPC-Cu 0.5 displayed significant cell death. Finally on day 7, CPC-Cu 0.1, 0.5, and 1 displayed similar hMSC viability to the blank CPC which indicates that the levels of Cu loaded in these scaffolds are in fact not toxic to the cells. It can be concluded that large amounts of Cu such as 3% or 5% are significantly toxic to hMSCs whereas lower concentrations of 0.1%, 0.5%, and 1% are ideal amounts as they retain the cell viability of the blank CPC.

#### 2.4.3. Alkaline Phosphatase Activity

Alkaline phosphatase (ALP) is an early osteogenic marker expressed in hMSCs which is being used to evaluate the osteogenic potential of scaffolds in vitro [34]. The ability of CPC for the osteoconduction and osteodifferentiation of hMSCs is well known [35]. The ALP activity was analyzed over a 14-day period. The data (Figure 5A) displayed no significant difference in ALP expression between the groups. This shows that the Cu loaded CPC scaffolds CPC-Cu 0.1, 0.5, and 1 have not shown any detrimental effect on the osteogenic differentiation of hMSCs compared to CPC. Further, scaffolds CPC-Cu 3 and CPC-Cu 5 displayed reduced potential which could be due to the low cell proliferation, as observed from cell proliferation studies (Figure 4B). These results indicate that Cu loaded scaffolds at lower Cu amounts have higher potential to improve hMSCs proliferation, as well as the differentiation of hMSCs to osteoblasts.

#### 2.4.4. Angiogenic Activity

Copper is well reported for its angiogenic potential [24]. Hence, it was pivotal to understand the angiogenic activity of the Cu loaded CPC scaffolds. From the brightfield images shown in Figure 5B, all the Cu-loaded scaffolds showed tube formation capabilities depicted by the formation of interconnections between the HUVEC cells which have been pointed out using the red arrows. All the scaffolds did show significant improvement from the control CPC scaffold, as there were no evident tube formations compared to the positive control. Further, it was seen that CPC-Cu 1 had the greatest number of interconnecting linear structures, further showing the importance of Cu for angiogenic activity. Although all the scaffolds did have interconnecting pores of a diameter ranging from 300–500 μm which is known for enhanced angiogenic activity, only the Cu loaded scaffolds showed angiogenic potential in the 18h period. This underscores the ability of copper to not only improve the number of interconnections, but also the ability to expedite the formation of these interconnections [27].

#### 2.4.5. Antibacterial Study

The ability of the Cu loaded scaffolds to be used as an antibacterial was also analyzed. Considering that *Staphylococcus aureus* is the primary bacteria responsible for osteomyelitis, it was chosen for this study [36]. All Cu loaded scaffolds observed a reduced bacterial colony count which is visualized in both SEM imaging (Figure 6A), colony formation on LB agar plates (Figure 6B), and the graph (Figure 6C). The colony count reduction was observed to be significant in all Cu loaded scaffolds as compared to the control. The colony count reduction was inversely proportional to the Cu loading content which depicts the ability for Cu to be eluted in sufficient quantities to minimize bacterial cell growth on the scaffolds. To further understand the effect of Cu on *S. aureus*, scaffolds used for the study were visualized using ESEM. Figure 6A shows the change in morphology of the bacterial cells on increasing exposure to Cu. This change in morphology is attributed to bacterial cell death and has been pointed out by the red arrows in the image [27]. It is interesting to observe that CPC-Cu 3 did in fact show the highest bacterial cell death and potentially indicates that the effect of copper plateaus at this concentration.

## 3. Discussion

Non-unions in bone often arise due to an underlying condition such as cancer or osteomyelitis. The regions hosting the non-union fractures often require external intervention to supplement the growth of new bone to heal the defect [37,38]. Considering that CPCs have widespread applicability, we attempted to improve the current state-of-the-art of ceramic CPC materials for bone regeneration. Copper ions are known for their ability to induce angiogenesis and antimicrobial activities [39]. The Cu-loaded CPC paste did significantly impede the printability with the addition of Cu NPs powder to the commercial CPC paste due to the altered powder-to-liquid ratio which is a crucial factor for the printability of a material in 3D printing [40]. Initial attempts to print these scaffolds were unsuccessful, even at printing pneumatic pressures as high as 600 kPa. Hence a novel liquid carrier Kollisolv MCT 70 was utilized to balance consistency and printability without impacting the mineralization capabilities of CPC. The composition of this liquid carrier was chosen in line with the CPC composition described in the patent where a medium chain glyceride, surfactant, as well as the emulsifier at a ratio of 15:3:2 was used [41]. These modifications proved successful in subsequent attempts to print at pressures ranging from 100–300 kPa. The high wettability or hydrophilic property of the scaffolds facilitate better cell attachment and cell proliferation. Contact angle goniometer measurements were conducted on the printed scaffolds to access the hydrophilicity of this material. The addition of Cu NPs to the CPC had significantly decreased the contact angle. This reduction in the contact angle is indicative of a more hydrophilic surface [42,43,44].

Previous studies have shown that the addition of Cu^2+^ has impeded the setting process/growth of HA. Cu has the potential to react with CPC powders and substitute the apatite which in turn retards the hydration of DCPA and further impedes setting [24]. The XRD results taken post 3 days in humidity of the printed scaffolds shows that all the Cu-loaded formulations retain the ability to form HA which shows that the strategy of mixing Cu NPs with CPC paste has not impacted the latter’s mineralization potential. The FTIR analysis showed that the addition of Cu NPs to the formulation did not cause any interactions. The release of copper showed a significant copper concentration dependent drug release for the porous CPC-Cu scaffolds.

Potentiating the osteogenic capability of these biomaterials through the addition of trace elements has been proven to be beneficial for repairing bone defects post tumor resection. Copper embodying one of these elements made it a suitable option [45]. It has also been proven that the adsorption of Cu^2+^ in calcium polyphosphate scaffolds promoted the adhesion and proliferation of osteoblast-like cells as well as improved ECM matrix production [46]. Cell attachment studies on the printed scaffolds indicated the ability of hMSCs to attach to the surface and it was observed that lower concentrations of copper i.e., CPC-Cu 0.1, 0.5, and 1 had the highest amount of cell attachment. Alkaline phosphatase is an enzyme present in the bone as well as the liver and its levels are often associated with the ability of osteoblasts to proliferate in the region [47]. ALP levels in scaffolds CPC-Cu 0.1, 0.5, and 1 were higher, which is in line with the MTT assay data, which further confirmed the improved bone regenerative capabilities of these materials.

The ability of Cu NPs to induce new blood vessel formation was imperative to its selection and use in these materials. Angiogenic potential studies were conducted to analyze the following. HUVECs present in all the Cu-loaded scaffolds demonstrated the significantly better ability of the HUVECs to form interconnections as compared to the control scaffolds. CPC-Cu 1 showed the highest potential, and this could be attributed to the amount of copper being released from these scaffolds. Research previously conducted has shown that dietary Cu^2+^ above 1 mg/mL has the potential to cause cell death in vitro and in vivo in adults [24]. Hence, the scaffolds should release low concentration of Cu which ultimately can be seen as the lower Cu-loaded scaffolds having shown potentiated ability for cell growth and attachment. Scaffolds CPC-Cu 3 and 5 were not suitable for any of these studies and CPC-Cu 1 seemed to be the highest concentration of Cu NPs that demonstrated improved cell growth capabilities.

Finally, Cu^2+^ ions have been previously used for their antibacterial activity [48]. Considering that osteomyelitis is a common precursor for non-union fractures, the ability of these scaffolds to prevent further infection was crucial. *Staphylococcus aureus*, responsible for osteomyelitis, was chosen as the bacteria for this study [36]. Antibacterial studies conducted on these scaffolds showed that with increasing concentration, there was increased *Staphylococcus aureus* cell death. All colony counting values of the Cu-loaded scaffolds demonstrated the significant inhibition of bacterial cell growth as compared to the blank scaffold. Further, the bacterial cell morphology of the bacteria attached to the scaffolds were visualized using ESEM and a change in the morphology could be observed in all the Cu-loaded scaffolds which was representative of cell death.

In this study, a scaffold capable of improving angiogenesis with antibacterial properties was manufactured. This supplementation of Cu NPs in these ceramic materials has demonstrated improved capability as compared to the current state of the art and has the potential to be used in bone regenerative medicine, especially in the case of non-union fractures. Although, there are studies regarding the addition of copper to bone regenerative scaffolds, this is the first paper showing the ability of this material to be printed as well as the first paper to show the ability of copper to be used with the pasty CPC formulation. Ultimately, the need to 3D print these materials is essential, as the inherent porosity of the material is insufficient for new blood vessel formation and hence 3D printing provides the capability to alter the structure of the material as well as gives the capability to print structures that mimic the defect site [1,49].

## 4. Materials and Methods

### 4.1. Materials

All materials used in this study were analytical grade. The materials were used as such without any further modifications.

### 4.2. Preparation of Copper Loaded CPC

The powder component of the pasty calcium phosphate scaffolds is based on bio-cement (CPC) manufactured by InnoTERE (Dresden, Germany) which was first described by Khairoun et al. in 1997. Briefly, it consists of 60% α-tricalcium phosphate (α-TCP) with a calcium-to-phosphate ratio of 1.45. The carrier liquid of the cement paste was prepared using a medium-chain triglyceride Miglyol 812 (Cäsar and Loretz GmBH, 40721 Hilden, Germany), a surfactant Polysorbate 80 (Merck, Darmstadt, Germany), and an emulsifier phosphoric acid monohexadecyl ester (Amphisol A, DSM Nutritional Products, Kaiseraugst, Switzerland) at a mixing ratio of 15:3:2 by weight [1]. The powder and carrier liquid are present in a 4:1 ratio by weight. The copper nanoparticles of 300 nm were procured from a commercial vendor (Sky Spring Nanomaterials Inc., Houston, TX, USA) and various weight percentages (wt%) of copper were added and mixed with CPC using a mortar and pestle. To retain the extrudability of the paste with copper nanoparticle loading and to keep the powder-to-liquid carrier ratio the same, a novel liquid carrier Kollisolv MCT 70 (BASF, Iselin, NJ, USA) was used. The composition was Kollisolv MCT 70, polysorbate 80, and Lutensit A-EP at a weight ratio of 15:3:2 (acid phosphoric ester of a fatty alcohol ethoxylate, BASF, USA).

### 4.3. Printing of Copper Loaded CPC

A pneumatic extrusion-based bioprinter (BIOX, Cellink, Gothenburg, Sweden) was used to 3D print CPC and copper loaded CPCs were mixed with different wt% (Table 1). Cylindrical scaffolds of 6 mm diameter and 0.5 mm height with 45% rectilinear infill were printed at a printing speed of 5mm/s using a 25-gauge needle. Further cylindrical scaffolds for release were of 8 mm diameter and 1.5 mm height with 25% to maintain pore diameter between 300–500 µm at the same printing speed using a 22-gauge needle. The print speed was 5 mm/s with pressure that varied from 100–300 kPa. The printed constructs were further set at 37 °C using 95% humidity in an incubator.

### 4.4. Scaffold Characterization

#### 4.4.1. Injectability

To analyze the extrudability of the copper loaded CPC, injectability was conducted using a texture analyzer (TA-XT2 analyzer, Texture Technologies Corp, Hamilton, FL, USA). The injectability was analyzed by monitoring the load utilized to extrude the material from a plastic syringe of 2 mm diameter. The syringe was mounted vertically. A crosshead at a speed of 1 mm/s was used for 5 mm. The force required to extrude the CPC was recorded.

#### 4.4.2. Microscopy Imaging

The images for the 3D printed scaffolds were captured using Dino-Lite Edge 3.0 (Dino-lite Digital Microscope, New Taipei City, Taiwan). The same images were used to determine the pore and strut diameter through ImageJ (v1.41, Bethesda, USA). The surface morphology of the scaffolds was analyzed using a scanning electron microscopy (SEM, Quanta 650ESEM, FEI, Hillsboro, OR, USA). The scaffolds after mineralization were air-dried. The scaffolds were sputter-coated with palladium-gold and proceeded with SEM imaging at 15 KV voltage. For elemental analysis, used to understand the surface presentation of copper on the scaffolds, the Bruker XFlash 6160 EDX (Bruker, Bremen, Germany) was employed at a voltage of 15 KeV. The elemental composition was captured and analyzed using ESPRIT LiveMap software (Bruker, Bremen, Germany).

#### 4.4.3. Contact Angle Measurement

To understand the surface properties of the CPC-Cu scaffolds, contact angle measurements were conducted. An FTA200 Dynamic Contact Angle Analyzer (First Ten Angstroms, NH, USA) was used for the analysis. A sessile drop analysis with a 30 G needle was conducted using deionized (DI) water. The images captured were analyzed using the FTA32 software (First Ten Angstroms, Portsmouth, NH, USA). A contact angle was established between the perimeter of a DI-water droplet and the CPC or CPC-Cu scaffold solid surface it rests on.

#### 4.4.4. Fourier Transform Infrared Spectroscopy (FTIR)

The intermolecular interactions between copper and CPC were investigated using FTIR spectroscopy (iS50 FT–IR) equipped with a SMART OMNI-Sampler (Nicolet, Thermo Fisher Scientific, Waltham, MA, USA). The scaffolds were powdered, and around 10–15 mg was used for FTIR analysis. The % transmittance was analyzed from 4000 to 400 cm^−1^, at a resolution of 4 cm^−1^ (64 scans per run) and the background was collected prior to every sample run. The spectra were analyzed using the OMNICTM series software, (Thermo Fisher Scientific, USA).

#### 4.4.5. Powder X-ray Diffraction (XRD)

To analyze the formation of HA after the setting reaction, each of the scaffold compositions was powdered and analyzed using XRD (MiniFlex, Rigaku Corporation, Xizhimenwai Ave, Japan). The powder was placed in the sample holder and loaded into the machine. The 2θ was set from 5 to 70 degrees with a scan speed of 2°/min and a scan step of 0.02 degrees. The voltage and current were set at 45 V and 15 mA with an ultimate resolution of 0.0025. The diffractogram was analyzed using SmartLab Studio II software (Rigaku, Xizhimenwai Ave, Japan).

#### 4.4.6. Copper Ion Release Study

The Cu loaded scaffolds were placed in phosphate buffer saline, pH 7.4 for the release and the temperature was maintained at 37 °C. These studies were conducted under static conditions and aliquots of 1 mL were collected on 1, 7, and 14 days. To maintain sink conditions, the withdrawn release media was replaced with fresh PBS. Copper released from the scaffold samples was diluted with 2% nitric acid and was analyzed using inductively coupled plasma mass spectroscopy.

### 4.5. In Vitro Studies

The human bone marrow derived mesenchymal stem cells (hMSCs) were procured from American Type Culture Collection (ATCC, Manassas, VA, USA). The cells were expanded in knockout Dulbecco’s modified Eagle’s medium (DMEM, Gibco, Billings, MT, USA) supplemented with 20% fetal bovine serum (FBS, Gibco, USA) and 1% penicillin-streptomycin (Gibco, Billings, MT, USA) and 50 µg/mL gentamycin.

The human vascular endothelial cells (HUVECs) were procured from Gibco as part of the angiogenesis kit supplemented with HUVEC basal medium and Gibco™ large vessel endothelial supplement. Geltrex™ LDEV-free reduced growth factor basement membrane matrix was also provided in the kit.

The cells were cultured at 37 °C in a 5% CO_2_ incubator (CellExpert C170, Eppendorf, Framingham, MA, USA). The cells were monitored and used for study when they reached 80% confluency. The hMSCs were trypsinized using trypsin 0.25% ethylene diamine tetracetic acid (EDTA). The activity of trypsin was neutralized with serum media and centrifuged (Eppendorf, Germany) at 500× *g* for 5 min. The cells were mixed with trypan blue (Gibco, USA) and counted in a CountessII (Life technologies, Carlsbad, CA, USA).

#### 4.5.1. Cell Attachment

The cell attachment study was carried out by seeding 50,000 cells per scaffold and incubating them for 24 h. The cells were fixed with 10% formaldehyde (Sigma Aldrich, Saint Louis, MO, USA) for 10 min. Scaffolds were gradually dehydrated using gradient ethanol (50, 70, 80, 90, 95 and 100% ethanol) and the scaffolds were stored overnight in 100% ethanol. Before SEM imaging, the scaffolds were completely submerged in hexamethyldisilazane (HMDS, Sigma Aldrich, Saint Louis, MO, USA) for 10 min. HMDS was removed, and scaffolds were dried in the fume hood for about 30 min. The scaffolds were then sputter coated with gold-palladium material and used for SEM imaging.

Fluorescence images were also taken for cell attachment. After 1 day of incubation of the HMSCs on the scaffolds, spent media was removed. Cells were fixed on the scaffolds for 30 min using 3.7% formaldehyde. Further, fixed cells were washed with PBS and 1 µg/mL Alexa Fluor phalloidin 488 dye was added to the cells and incubated in the dark for 45 min at room temperature. After removing Alexa Fluor phalloidin 488 dye, 0.1 µg/mL DAPI was added, and the scaffolds were incubated for 5 min. HMSCs on the scaffolds were imaged under a fluorescent microscope equipped with blue and green channels.

#### 4.5.2. Cell Proliferation

To assess cell proliferation upon exposure to the copper loaded scaffolds, an MTT (3-(4, 5dimethylthiazol-2-yl)-2, 5-diphenyltetrazolium bromide, Sigma-Aldrich, Saint Louis, MO, USA) assay was performed. The scaffolds of 6 × 6 × 1 mm (length × width × height) cylindrical scaffolds were sterilized with ethanol for 15 min and then dried under ultraviolet light for 20 min and placed into a sterile 96 well plate. The hMSCs of 2 × 10^4^ cells per scaffold were seeded and cultured in the CO_2_ incubator at 37 °C for 1, 3, or 7 days. In the post-incubation period, the growth medium was removed and replaced with 10% MTT solution mixed with serum media. The plates were then incubated at 37 °C for 2 h. After this period, media was removed and 200 µL of DMSO was added to the plate. After 5 min of incubation at 37 °C, the dimethylsufoxide (DMSO, Sigma-Aldrich, Saint Louis, MO, USA) was added and read at 570 and 700 nm in a multi-well plate reader (BioTek, Winooski, VT, USA).

#### 4.5.3. Alkaline Phosphatase Activity

Alkaline phosphatase is an early bone marker establishing the differentiation potential of hMSCs towards osteoblasts. The hMSCs of 2 × 10^4^ cells were seeded on to the sterilized scaffolds kept in a 96 well plate. The ALP assay kit was used, and the experiment was conducted as per the manufacturer’s protocol. For the quantitative measurement of alkaline phosphatase expression on day 14, p-nitrophenyl phosphate (pNPP, Sigma Aldrich, Saint Louis, MO, USA) substrate was used. To lyse the cells, 0.1% Triton X-100 was used and the cell lysate was freeze-thawed for 10 min from −80° to 37 °C. An amount of 0.1 mL of the lysate was taken in a 96 well plate and mixed with an equal amount of p-NPP. The absorbance was recorded using a microplate reader at 405 nm after 1 h incubation.

#### 4.5.4. Angiogenic Activity

To assess the ability of these scaffolds to induce blood vessel formation, a tube formation angiogenesis assay was conducted. The experiment was conducted using an angiogenesis kit procured from Gibco, USA as per the manufacturer’s protocol. Briefly, human umbilical vein endothelial cells (HUVEC) were cultured in a 96-well plate at a seeding density of 2.5 × 10^3^ viable cells/cm^2^ and supplemented with Gibco™ large vessel endothelial supplement (LVES). The cells were then incubated at 37 °C in a humidified atmosphere of 5% CO_2_. On the day of the tube formation assay, Geltrex™ LDEV-free reduced growth factor basement membrane matrix was prepared by placing it into the desired wells. The basement membrane matrix was then seeded with 25,000 cells per cm^2^. Finally, supernatants from the scaffold wells were transferred to the matrix wells for the study. Wells containing Gibco™ large vessel endothelial supplement were used as the positive control and wells with only basement membrane matrix were the negative control. The angiogenic potential was then observed. The brightfield images were taken after 18 h of incubation using an inverted fluorescence microscope (IX-83, Olympus, Center Valley, PA, USA) equipped with an ORCA-flash 4.0 camera.

### 4.6. Antibacterial Study

The pathogenic bacteria *Staphylococcus aureus* responsible for osteomyelitis was used to assess the anti-bacterial activity of the CPC-Cu scaffolds. The bacteria were grown in Luria broth at 37 °C. The CPC scaffolds with and without Cu were kept in a 48 well plate with 0.1 mL of bacterial suspension normalized to 0.5 McFarland standard (1 × 10^8^ CFU/mL). After 24 h incubation, the bacterial suspension was diluted to 1 × 10^6^ CFU/mL with sterile phosphate buffer saline (PBS) (Gibco, USA). The 50 µL of the diluted bacterial suspension was streaked across an agar plate using a sterile swab and incubated overnight at 37 °C to facilitate colony formation. The formed bacterial colonies were counted and reported. Further, the scaffolds with bacteria attached to them were washed with PBS and fixed with 3.7% formaldehyde. The scaffolds were treated with gradient ethanol (50, 70, 80, 90, 95, and 100%) for 10 min each. Finally, the scaffolds were treated with HMDS and air-dried. The scaffolds were coated with palladium-gold and visualized using SEM (Quanta 650ESEM, FEI, USA).

### 4.7. Statistics

All samples were run in triplicates. The data are represented as mean ± standard deviation. The significance of the difference *p* < 0.05 was evaluated using one-way or two-way ANOVA with post Bonferroni or Tukey tests using GraphPad (Prism9, La Jolla, CA, USA).

## 5. Conclusions

The effects of adding Cu NPs with angiogenic and antibacterial capabilities to traditional calcium phosphate cement scaffolds were studied. The results demonstrated that CPC-Cu 0.1, 0.5 as well 1 showed improved capabilities in terms of hMSC cell attachment, viability, and bacterial cell death. However, the addition of Cu NPs higher than 1% showed reduced potential to be used as a biomaterial, as it led to significant hMSC cell death. CPC-Cu 1 scaffolds showed improved angiogenic and antibacterial potential compared to other groups. In conclusion, Cu NPs addition is a suitable carrier for Cu^2+^ ions which shows promise as a biomaterial that can be used for new blood vessel formation and in preventing osteomyelitis in non-union fractures.

## Figures and Tables

**Figure 1 pharmaceuticals-16-00352-f001:**
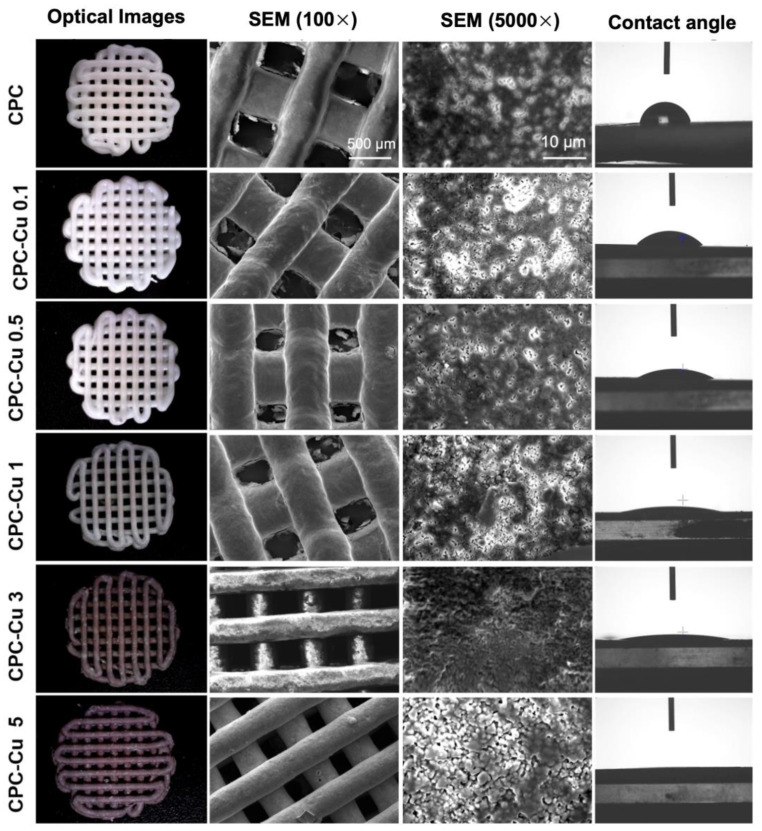
The characterization of as printed scaffolds imaged under optical microscopy (left panel 1) scanning electron microscopy images at 100× and 5000× showing the top surface morphology of the scaffolds (panel 2 and 3) and wettability of the scaffolds with images taken using a contact angle goniometer showing the change in contact angle lower with Cu loaded CPC scaffolds compared to CPC.

**Figure 2 pharmaceuticals-16-00352-f002:**
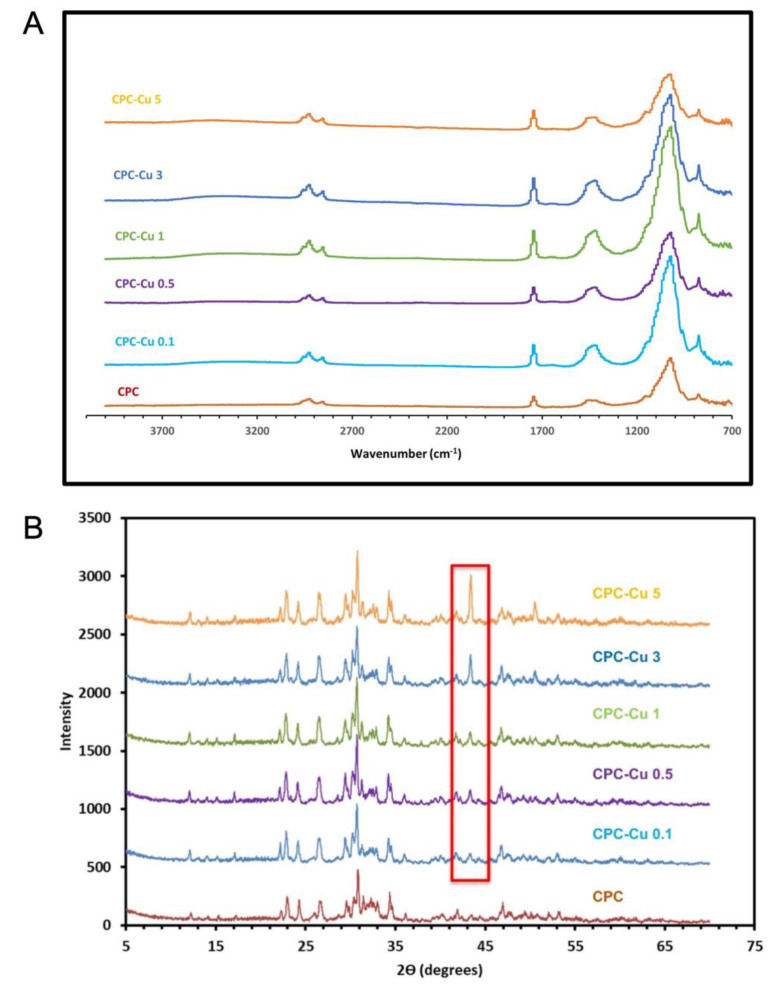
Characterization of scaffolds to understand the surface chemistry with (**A**) FTIR and crystallinity using (**B**) X-ray diffraction spectrum where the part marked in the red box is indicative of the copper peaks.

**Figure 3 pharmaceuticals-16-00352-f003:**
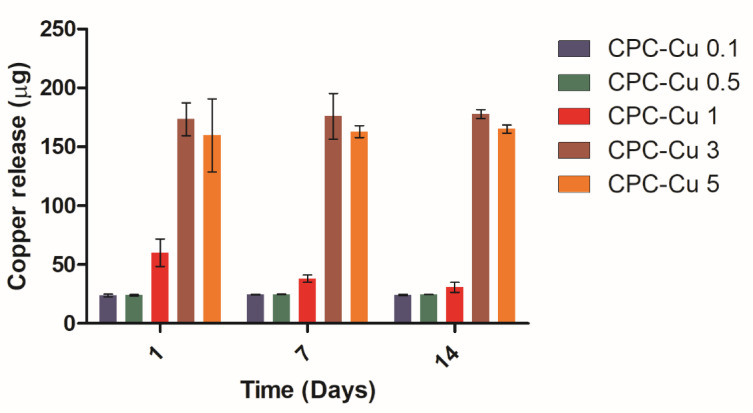
The plot shows the copper release in 14 days from CPC-Cu scaffolds in PBS at pH 7.4. The data are represented as mean ± standard deviation for *n* = 3.

**Figure 4 pharmaceuticals-16-00352-f004:**
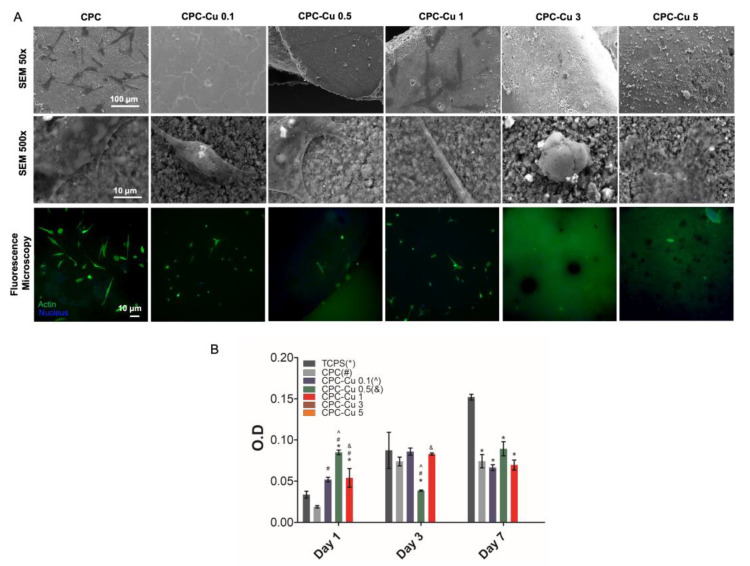
in vitro studies showing (**A**) cell attachment of hMSCs cells after day 1 incubation on CPC and Cu loaded CPC scaffolds showing changes in cell morphology with increased copper amounts in the scaffolds imaged under low mag SEM, high mag SEM, as well as fluorescence microscopy and (**B**) cell proliferation of hMSCs for day 1, 3, and 7 durations on CPC and Cu loaded CPC scaffolds showing improved cell proliferation with respect to time and no cell proliferation on scaffolds CPC-Cu 3 and CPC-Cu 5 groups. The data are represented mean ± standard deviation (*n* = 3). The significance of difference *p* < 0.05 for various groups.

**Figure 5 pharmaceuticals-16-00352-f005:**
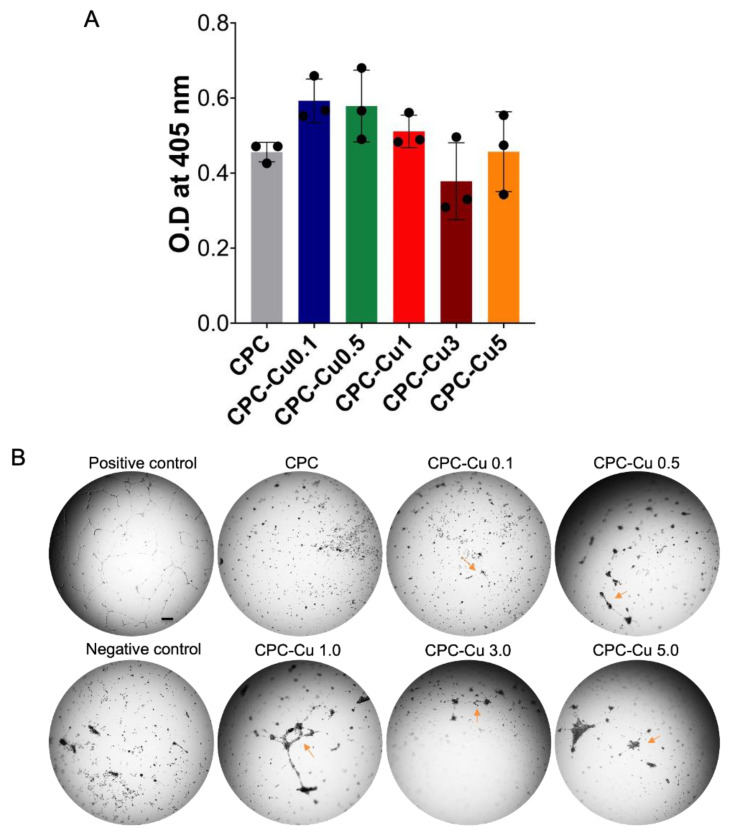
The multifunctionality of Cu loaded CPC scaffolds showing (**A**) alkaline phosphatase activity. The data are represented by mean ± standard deviation (*n* = 3) and there is no significant difference between the groups and (**B**) angiogenic potential with arrows showing the tube formation on the extracellular matrix. The CPC-Cu 1 scaffolds showed increased tube formation compared to other Cu loaded CPC scaffolds.

**Figure 6 pharmaceuticals-16-00352-f006:**
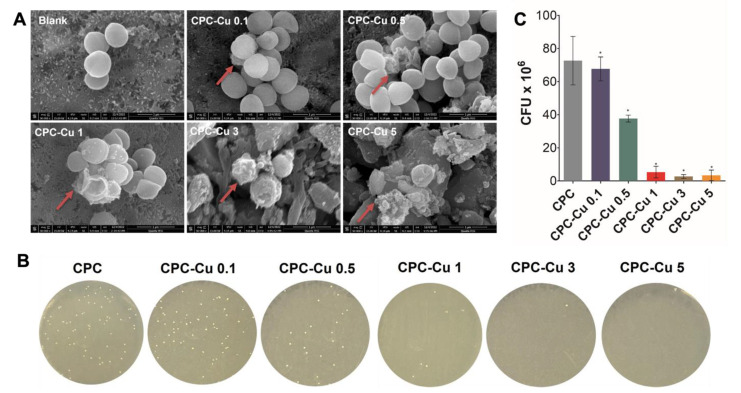
The antimicrobial activity of CPC and Cu loaded CPC scaffolds; (**A**) scanning electron microscopy images showing the deformed *S. aureus* morphology; (**B**) quantified *S. aureus* bacteria; and (**C**) colony forming units (CFU) of *S. aureus* cultured on Luria agar plates after treatment for 24 h. The data are represented as mean ± standard deviation (*n* = 3). The significance of difference * *p* < 0.05 for various groups compared to CPC.

**Table 1 pharmaceuticals-16-00352-t001:** Formulation and preliminary characterization of CPC and copper loaded CPC at different wt%.

Sample	CPC (wt%)	Cu-NP (wt%)	Strut Diameter (µm)	Pore Diameter (µm)	Extrusion Peak Force (g)	Contact Angle (°)
CPC	100	0	434 ± 25	434 ± 17	3090 ± 41	75 ± 9
CPC-Cu 0.1	99.9	0.1	457 ± 38	433 ± 20	4018 ± 37	35 ± 9
CPC-Cu 0.5	99.5	0.5	462 ± 15	466 ± 28	4292 ± 137	9 ± 2
CPC-Cu 1	99	1	467 ± 22	457 ± 13	3675 ± 101	10 ± 2
CPC-Cu 3	97	3	453 ± 28	453 ± 24	4524 ± 150	8 ± 2
CPC-Cu 5	95	5	451 ± 12	451 ± 13	3615 ± 56	3 ± 1

## Data Availability

The data presented in this study are available on request from the corresponding author. The data are not publicly available due to confidentiality.

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
