# Peer review of "Multifunctional Three-Dimensional Printed Copper Loaded Calcium Phosphate Scaffolds for Bone Regeneration"

_pharmaceuticals, 2023, doi:10.3390/ph16030352_

Round 1

Reviewer 1 Report

The manuscript “Multifunctional three-dimensional printed copper loaded calcium phosphate scaffolds for bone regeneration” by Amit Pillai et al. devoted on new materials for bone regeneration. The study presents extensive experiments, using a number of methods. After careful consideration I recommend minor revision. However, I have some comment and questions, that are listed below.

1.       The references are not meet MDPI requirements: the first there must be ref., and after the dot.

2.       The proportion of the liquid phase (Kollisolv MCT 70, Polysorbate 80 and Lutensity A-EP) to Cu-NP should be mentioned.

3.       The value of the scale bar must be shown in the figure 1.

4.       Fig. 2 (XRD patterns): the authors should include standard PDF card for HA phase for comparison at the bottom. Also, all copper peaks must be marked in Fig. 2.

5.       Why the copper release for the scaffold samples is not changing with the duration of the soaking?

6.       Some important works on Cu-doped calcium phosphate materials should be cited:

·         Materials Science & Engineering C 99 (2019) 1199–1212 doi.org/10.1016/j.msec.2019.02.042;

·         Ceramics International, Volume 48, Issue 20, 2022, Pages 29770-29781, doi.org/10.1016/j.ceramint.2022.06.237

·         SSRN Electronic Journal, 2022, doi.org/10.2139/ssrn.4295801

Some typos:

In the Funding selection remove “Please add:”

The References must be updated according to MDPI requirements.

Author Response

Reviewer 1:

  1. The references are not meet MDPI requirements: the first there must be ref., and after the dot.

The authors understand the reviewer’s concern and have made changes to the references. The references have been changed to Pharmaceuticals Journal style.

  1. The proportion of the liquid phase (Kollisolv MCT 70, Polysorbate 80 and Lutensity A-EP) to Cu-NP should be mentioned.

The authors understand the reviewer’s concern and address it by adding the weight ratio to the manuscript. The composition was Kollisolv MCT 70, polysorbate 80 and Lutensit A-EP (Acid phosphoric ester of a fatty alcohol ethoxylate, BASF, USA) at a weight ratio of 15:3:2.

  1. The value of the scale bar must be shown in the figure 1.

The images have been modified as per the reviewer’s comments and scale bars with the values have been added to the images.

  1. Fig. 2 (XRD patterns): the authors should include standard PDF card for HA phase for comparison at the bottom. Also, all copper peaks must be marked in Fig. 2.

The authors understand the concern put forth by the reviewer. The XRD data for HA was cross checked with the JCPDS database where the data fit to HA with the card number 9-0432. The same was also checked with literature (Cryst. Res. Technol., 1–8 (2015) / DOI 10.1002/crat.201500143). The copper peaks were marked with a red color box.

  1. Why the copper release for the scaffold samples is not changing with the duration of the soaking?

The authors understand the reviewer’s comments. This is due to the burst release of copper from the Cu loaded scaffolds. The copper is released by diffusion and hence most of the copper is released within the first day. The rest which is slowly released over the duration of the experiment.

  1. Some important works on Cu-doped calcium phosphate materials should be cited:
  • Materials Science & Engineering C 99 (2019) 1199–1212 doi.org/10.1016/j.msec.2019.02.042;
  • Ceramics International, Volume 48, Issue 20, 2022, Pages 29770-29781, doi.org/10.1016/j.ceramint.2022.06.237
  • SSRN Electronic Journal, 2022, doi.org/10.2139/ssrn.4295801

The following research articles have been cited taking to regard the reviewer’s comment.

Some typos:

  1. In the Funding selection remove “Please add:”

The authors have removed the “please add.” in the funding section. Further typos have been addressed as well.

  1. The References must be updated according to MDPI requirements.

The references have been changed to the required format. The references have been changed to the pharmaceuticals format.

Reviewer 2 Report

The research article entitled “Multifunctional three-dimensional printed copper loaded calcium phosphate scaffolds for bone regeneration” is study of 3D printed copper nanoparticle loaded calcium phosphate cement scaffolds for bone regeneration.

Article has good merit but some overstatements can be revised. There are some points to be addressed are given below;

-       CuNP synthesis should either be detailed or referred to literature. Additionally, NP data should be provided under supplementary information.

-       Page 3, line 112; It is mentioned that the contact angle of all CPC-Cu scaffolds has contact angle ≤ 15°. However, the CPC-Cu 0.1 scaffold has 35 ° contact angle. Rephrase is needed.

-       Spectrums at Figure 2 can be enlarged. Not easy to read.

-       Line 148 “However, as the Cu content increases, the crystallinity of the prepared cement is also seen to increase.” statement is not clear in the spectrum. Please detail the statement or remove it.

-       It’s not convincing that there is any cell attachment on the surface from Figure 4A. The arrow indicated structures are too big to be cell (100 um) and the number is very low. There is no polypod structure of the cells that is the most prominent form when they attach to a surface. Also, it is unrealistic to expect a cell morphology change in a day after cell seeding. These data should be repeated or removed.

-       Figure 4.B. Negative absorbance values might be the result of assay conditions. Better to give these values as 0 (zero) instead of negative.

-       Figure 5.B image qualities should be improved.

-       Based on ALP data, it is not possible to conclude that Cu addition enhanced osteogenesis. This statement should be revised throughout the manuscript especially under conclusion.

Author Response

Reviewer 2:

  1. CuNP synthesis should either be detailed or referred to literature. Additionally, NP data should be provided under supplementary information.

Authors appreciate the reviewers comment. The nanoparticles were procured from a commercial vendor and they are not in a situation to disclose the manufacturing process. However, the authors provided the SEM image of the copper nanoparticles in supplementary information (Figure S1).

  1. Page 3, line 112; It is mentioned that the contact angle of all CPC-Cu scaffolds has contact angle ≤ 15°. However, the CPC-Cu 0.1 scaffold has 35 ° contact angle. Rephrase is needed.

The authors understand the reviewer’s concern. The sentence has been rephrased to describe that the contact angle below 15 can be seen in all the scaffolds except CPC-Cu-0.1.

  1. Spectrums at Figure 2 can be enlarged. Not easy to read.

The image has been enlarged as per the reviewers’ comment.  

  1. Line 148 “However, as the Cu content increases, the crystallinity of the prepared cement is also seen to increase.” statement is not clear in the spectrum. Please detail the statement or remove it.

The sentence has been removed as per the recommendation of the reviewer.

  1. It’s not convincing that there is any cell attachment on the surface from Figure 4A. The arrow indicated structures are too big to be cell (100 um) and the number is very low. There is no polypod structure of the cells that is the most prominent form when they attach to a surface. Also, it is unrealistic to expect a cell morphology change in a day after cell seeding. These data should be repeated or removed.

The authors understand the reviewer’s concern. The size of a typical stem cell is 7-10 mm when the cell is typsinized and circular. However, on surface, the stem cells grow to more than 100 micrometers length (https://doi.org/10.1016/j.matdes.2018.11.018). The objective of cell attachment is to verify the ability of cells to adhere to the surface due to surface morphology and surface chemistry. In our experiment, the cells were attached and spread on the surface of control and low cu wt% scaffolds. However, they were circular and didn’t attach or spread in higher cu wt % scaffolds due to the changes in surface chemistry (higher cu ions) where the cells might undergo stress. We checked the data thoroughly and we concluded to keep the data in the manuscript.

  1. Figure 4.B. Negative absorbance values might be the result of assay conditions. Better to give these values as 0 (zero) instead of negative.

    The negative values were a byproduct of subtracting the blank from the values. The authors understand the concern put forward by the reviewer and have changed the image to show all negative values as zero.

    1. Figure 5.B image qualities should be improved.

    The qualities of the image have been improved to clearly show the angiogenic potential in the images.

    1. Based on ALP data, it is not possible to conclude that Cu addition enhanced osteogenesis. This statement should be revised throughout the manuscript especially under conclusion.

    The authors understand the concerns of the reviewers. The sentence has been changed in all circumstances where this statement was used. Further the heading osteogenic activity has been changed to alkaline phosphatase activity.  

Round 2

Reviewer 2 Report

The research article entitled “Multifunctional three-dimensional printed copper loaded calcium phosphate scaffolds for bone regeneration” is study of 3D printed copper nanoparticle loaded calcium phosphate cement scaffolds for bone regeneration.

Article is improved in the revised manuscript with one exception. Despite the authors explanation, it’s not convincing that there is any cell attachment on the surface from Figure 4A. The arrow indicated structures are too big to be cell (100 um) and the number is very low. There is no polypod structure of the cells that is the most prominent form when they attach to a surface. Also, it is unrealistic to expect a cell morphology change in a day after cell seeding. Authors might provide more data in supplementary information to convince the reader that their statement is valid. Also showing the cells grown on control cover slips to indicate the resemblance. Outherwise, the data should be removed.

Author Response

Authors Response to Reviewer

The authors appreciate the reviewers time and effort in reviewing our research. We have obliged with the reviewer’s comments and provided the response below.

Reviewer 1:

  1. Article is improved in the revised manuscript with one exception. Despite the authors explanation, it’s not convincing that there is any cell attachment on the surface from Figure 4A. The arrow indicated structures are too big to be cell (100 um) and the number is very low. There is no polypod structure of the cells that is the most prominent form when they attach to a surface. Also, it is unrealistic to expect a cell morphology change in a day after cell seeding. Authors might provide more data in supplementary information to convince the reader that their statement is valid. Also showing the cells grown on control cover slips to indicate the resemblance. Outherwise, the data should be removed.

We thank reviewer for his appreciation of the revised manuscript. We welcome reviewer’s valuable comment on the cell attachment. To alleviate the suggested concerns, we took an effort to redo the experiment. Now we have included high magnification SEM images of the cell showing the polypod structures with the cell surface. Besides, we also included fluorescence microscopy images in figure 4 to show the hMSC cell size and shape for better understanding. The incorporated change was also provided here below.

Figure 4

Round 3

Reviewer 2 Report

It was very nice to add more images to Figure 4!